# Simplified Uncertainty Bounding: An Approach for Estimating Flood Hazard Uncertainty

**Tim Stephens** [1,2,*] **and Brian Bledsoe** [2]

1 Dynamic Solutions, LLC, Knoxville, TN 37919, USA
2 Institute for Resilient Infrastructure Systems, University of Georgia, Athens, GA 30602, USA; bbledsoe@uga.edu
* Correspondence: tastephens@dsllc.com

**Abstract:** Deterministic flood hazard estimates neglect the inherent uncertainty associated with model estimates and can substantially underestimate flood risk. Monte Carlo simulation (MCS) has been a valuable tool for conducting uncertainty analysis. However, its application has primarily been limited to a single research setting. Recent development of a point approximation method, simplified uncertainty bounding (SUB), simulated the uncertainty from MCS with high accuracy (e.g., a critical success index of 0.75). However, an evaluation of additional flood hazard metrics and hydro-climate settings that impact the distribution of uncertainty is required. We evaluated SUB at two contrasting study sites by comparing their results with MCS and identified scenarios where performance increased and decreased. The SUB method accurately matched aerial inundation metrics, but performance was reduced for relative errors in flood depth and top width. Hydraulic structures had a heterogeneous impact on accuracy, and the confinement ratio had a positive relationship with the top width error. While SUB generally performed well with relative errors of approximately ±10% for a 90% confidence interval, some outliers did exist. The acceptability of the approach will depend on the specific application. Though SUB overestimated uncertainty, it provides a conservative estimate and is a cost-effective alternative to MCS.

**Keywords:** flood modeling and inundation mapping; uncertainty analysis in flood hazard layer development; fluvial flood hazard; probabilistic floodplain mapping



## 1. Introduction

As global flood losses continue to rise, evidence has emerged that highlights the potential impacts of uncertainty on flood risk estimates and the need to incorporate uncertainty into floodplain management [1–4]. Despite a significant portion of flood insurance claims occurring outside regulatory flood hazard boundaries [5,6], standard methods continue to depict flood hazards as deterministic estimates that convey a precisely known outcome. Considering the density of infrastructure and development patterns immediately adjacent to deterministic flood hazard boundaries [7,8], quantifying the uncertainty in flood extent could identify locations and assets with elevated flood risk.

Sources of uncertainty in model estimates of flood hazards are numerous and well documented [9,10]. Monte Carlo simulation (MCS) is a useful tool to estimate uncertainty in flood hazard estimates, but this methodology has primarily been limited to a single research setting. Alternative methods to approximate uncertainty, such as first order approximation, require knowledge of derived response functions that are difficult to determine for complex hydraulic models. A cost-effective alternative to quantify uncertainty in flood hazards estimates from hydraulic models could be a valuable tool in floodplain management.

Stephens and Bledsoe [3] introduced a potential method that matched the uncertainty in inundation areas estimated from MCS with high accuracy. Their approach, hereafter referred to as simplified uncertainty bounding (SUB), implemented systematic

sampling and can be classified as a point estimation method. However, SUB requires further testing of additional flood hazard metrics at locations with contrasting climates and geomorphic settings.

Historic and projected increases in flood loss present a challenge for floodplain management and underscore the potential value in quantifying and transparently portraying uncertainty in flood hazard estimates. While multiple methods exist for quantifying this uncertainty, quantifying differences and influential factors among approaches that vary in complexity is required to more clearly understand their limitations and increase applicability across a broader audience.

This study seeks to determine how flood hazard uncertainty estimated by an alternative, cost-effective approach compares to MCS. It also seeks to identify factors, in addition to model inputs and parameters, that influence uncertainty estimates. We conducted the MCS and SUB of 1-dimensional hydraulics at multiple locations with contrasting climatic and geomorphic settings. The SUB method was evaluated by quantifying errors in flood depth, top width, and inundated area relative to MCS. Potential influences from factors in addition to stochastic forcing, such as geomorphic and hydraulic metrics, were evaluated through regression analysis with the error of SUB. Despite its cost-effectiveness, the application of SUB in the literature is limited, and this is the first study that evaluates its performance compared to MCS for regulatory floodplain models at contrasting settings.

Sections 1.1–1.3 provide a more detailed background on the uncertainty of flood hazard estimates by 1-D hydraulic models, various methods for approximating that uncertainty, and the influence of hydraulic geometry on uncertainty propagation. Section 2 describes the study locations of this analysis, the hydraulic model implemented, methods for estimating uncertainty in flood hazard estimates, and methods and metrics for evaluating factors that influence uncertainty propagation. Section 3 presents the results, followed by a discussion of those results in Section 4 and conclusions in Section 5.

### 1.1. Uncertainty in Hydraulic Model Results

Uncertainties in hydraulic modeling stem from a lack of knowledge and random variability [11]. Random variability can typically be defined statistically, but knowledge deficiencies may be difficult to quantify statistically due to changing characteristics in time and space. The stochastic occurrence of flood events provides an example of random variability, and the subjective assignment of friction parameters in hydraulic models to represent roughness that is spatially and temporally variable is an example of uncertainty from knowledge deficiencies. For practical purposes, uncertainties from knowledge deficiencies are often treated as though they are random in nature, and transparency has been suggested in assumptions about uncertainties in hydraulic modeling and their implications for flood hazard estimates [12,13].

While flood hazard estimates contain numerous sources of uncertainty, discharge, friction parameters, and topography have been consistently documented as the most influential sources within a particular modeling scheme [10,14]. However, the model structure or numerical scheme itself can be a significant source of uncertainty [15,16].

### 1.2. Approximation Methods

Numerous methods, such as MCS and first order approximation (FOA), have been apply to quantify uncertainty in open channel flow conditions [17–20]. While MCS directly quantifies the distribution of results through a large number of simulations, FOA provides an approximation by estimating the mean and variance of results [21]. However, the application of FOA requires limiting assumptions, such as near normal distributions of uncertain inputs, linear models, and a relatively small coefficient of variations in uncertain inputs [21], whereas other techniques such as MCS are computationally intensive. Other uncertainty approximation techniques include point estimation methods, and they approximate uncertainty by quantifying specific points of a result distribution [22]. Applications of point estimation methods to flood hydraulics is lacking.

MCS of a finite difference model was employed to obtain distributions of flow variables along a reach of the Columbia River based on a description of the spatiotemporal uncertainty in model parameters derived from comprehensive field data [19]. The results of that analysis indicated that the distribution of results was non-homogeneous in space and time, and the relative impact of model parameters on flow variables differed. For instance, velocity was most sensitive to cross-section geometry, whereas bed slope had the largest impact on flow depth. Numerous other applications have implemented MCS in various frameworks to evaluate uncertainty in hydraulic model estimates [1,23–32].

Less computationally intensive approaches have implemented FOA with Manning's equation to probabilistically describe flow variables as a result of parameter uncertainty [17,18]. These studies considered friction slope and roughness as the predominant sources of uncertainty, but they did not evaluate the uncertainty from FOA against other estimation techniques such as MCS. However, applications with water quality models have shown that FOA can provide satisfactory estimates for the central tendency of a distribution when compared against MCS, but the results between the two approaches diverged in the tails of the distribution [33]. For non-linear systems, FOA becomes less accurate as the parameter values depart from their mean [34].

Tyagi and Haan [20] developed a method to correct FOA errors for power function models, and they documented the magnitude of errors for various model exponents. They show that FOA errors increase with increasing uncertainty in model inputs and increasing non-linearity in model structure. However, the application of this approach to standard hydraulic models may not be viable due to the complexity of the model structure. For example, Tyagi and Haan [20] provided an example using Manning's equation, but common hydraulic models implemented to delineate flood hazards tend to become more numerically complex as the space–time dimensions of the model domain increase in complexity. The authors noted that, in more complex models, MCS may provide a more practical approach and that it is difficult to predict the error based on input and parameter distributions alone due to the complex functional form of hydraulic models.

An application of FOA with HEC-RAS to estimate parameter uncertainty propagation exemplifies the complications of FOA when the mathematical response function is difficult to deduce a priori [29]. In that particular example, MCS was used to quantify the response function and evaluate the relationship between inundation and topography, discharge, and friction parameter uncertainty. Their analysis revealed a complex relationship in inundation uncertainty between topography, discharge, and friction parameters. Further, the nature of this relationship varied among two reaches with different valley types.

The SUB method is a point estimation technique that requires a reduced number of model simulations based on a systematic sampling of uncertain model inputs and parameters. It has been shown to approximate the range of uncertainty in the inundation extents derived from MCS with a high degree of accuracy, providing an attractive alternative [3]. However, the method has only been tested at a single location, and its transferability to other sites is unknown due to variability in factors that might influence the performance of the approach. For instance, spatial variance in flood hazard uncertainty suggested that a local valley form might impact uncertainty distributions. Further, performance was evaluated based solely on the accuracy of inundated area. Discrepancies in water surface elevation or flood depth might also be important and not represented by measures of fit based on inundated area.

### 1.3. Influence of Hydraulic Geometry

The Saint-Venant equations for a gradually varied, unsteady 1-dimensional (x-direction) open channel flow can be expressed through the conservation of mass as [35]

$$\frac{\partial(AV)}{\partial x} + \frac{\partial A}{\partial t} - q = 0 \tag{1}$$

and the conservation of linear momentum can be expressed as

$$\frac{\partial V}{\partial t} + V\frac{\partial V}{\partial x} + g\left(\frac{\partial h}{\partial x} - S_0 + S_f\right) = 0 \tag{2}$$

where $A$ is the cross-sectional area, $V$ is the velocity, $t$ is the time, $x$ is the longitudinal distance along the channel, $q$ is the lateral inflow or outflow, $h$ is the flow depth, $S_0$ is the channel bed slope, $S_f$ is the friction slope, and $g$ is the acceleration due to gravity. The friction slope is typically solved by an empirical roughness relation such as Manning's equation:

$$S_f = \frac{\bar{u}^2 n^2}{R^{4/3}} \tag{3}$$

where $n$ is Manning's roughness coefficient, and $R$ is the hydraulic radius. Equations (1) and (2) contain three independent variables: $x$, $t$, and $S_0$; they contain two dependent variables: $h$ and $V$. Area and friction slope are functions of flow depth. For irregular channels, empirically derived power functions can define the relationship between hydraulic geometry parameters (e.g., area and hydraulic radius) and flow depth, for example [36],

$$A = ah^b + \varepsilon_A \tag{4}$$

where $a$ and $b$ are cross-sectionally unique parameters of the power function, and $\varepsilon_A$ represents an error term. Thus, the solution to depth and velocity will be partially dependent on the relationship between depth and hydraulic geometry parameters, which can vary in space along a reach. Further, Equation (2) highlights the fact that the solution is also partially dependent on adjacent hydraulic controls, such as expansions or contractions that might be a result of bridges, a valley form, etc.

Longitudinal variability in hydraulic geometry parameters will result in longitudinally varying sensitivity of uncertainty depending on the metric in question. Consequently, the error in uncertainty approximation methods is expected to differ as a result of varying sensitivities. For instance, the rate of change in water surface elevation corresponding to a unit change of inputs and parameters will be different for a v-shaped valley compared to a u-shaped valley, and the resulting change in flow depth versus inundated area will be different among the two locations.

While this relationship can be shown conceptually using the governing equations of fluid dynamics, it is difficult to evaluate analytically for practical applications due to complex hydraulic interactions between downstream/upstream boundary conditions, friction parameters, and local topography. Therefore, numerical modeling experiments present an attractive approach to evaluate the performance of approximate methods for uncertainty analysis. The intention here is not to fully identify the causal mechanisms describing errors in uncertainty approximation methods. Rather, we have provided this conceptual illustration to guide the selection of commonly computed hydraulic variables that might correlate with the relative magnitude of errors and indicate scenarios of reduced accuracy.

## 2. Materials and Methods

### 2.1. Study Sites

Bronx Wash, Tucson, AZ, USA, and Proctor Creek, Atlanta, GA, USA, served as study sites to evaluate SUB against MCS and investigate factors that might increase/decrease its accuracy (Figure 1). Conducting model experiments with study sites provided an advantage over hypothetical channel geometries by including complexities of urban environments, such as a heterogeneous topography, in-stream structures (e.g., bridges and culverts), and their simultaneous impacts on hydraulics. The two study sites were selected based on fundamental differences in the climate, hydrologic regime, urban watershed structure, geomorphic context, and socio-political environment to account for a wide range of influences on uncertainty distributions. However, both watersheds are highly urban (Table 1).

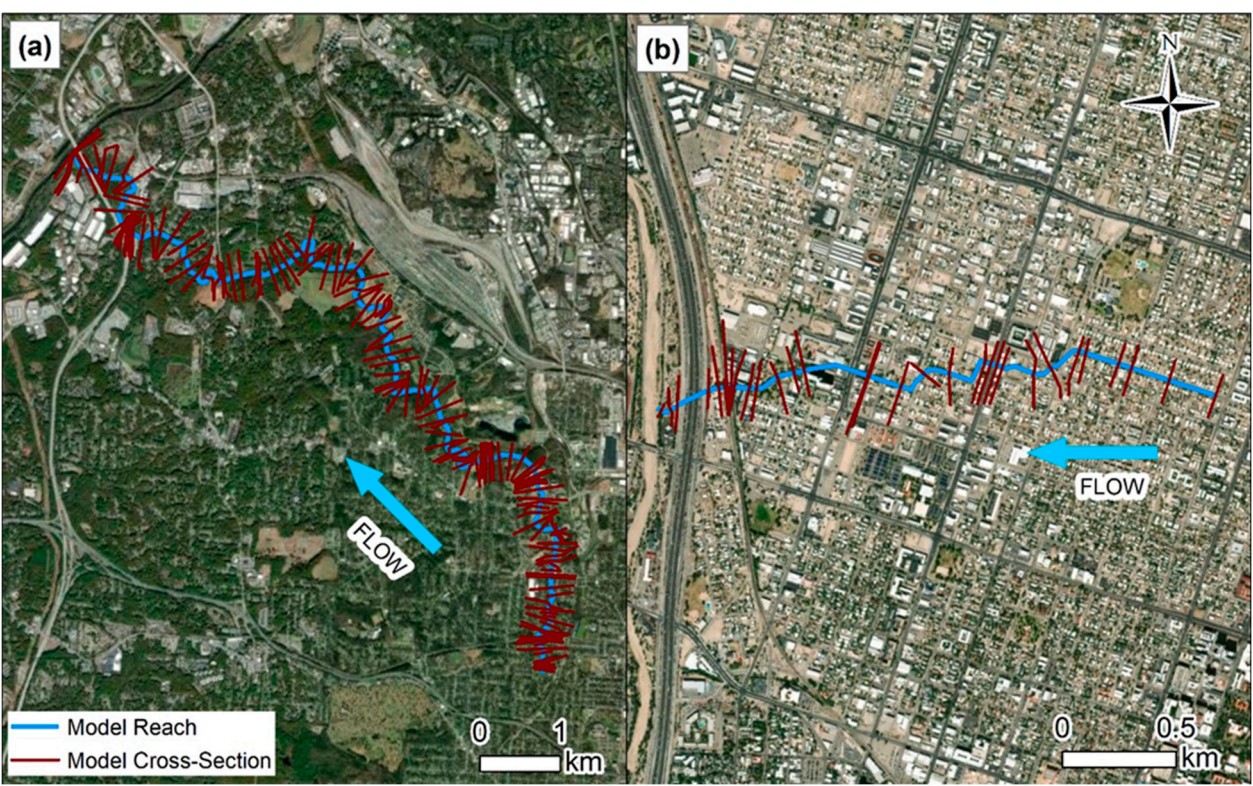

**Figure 1.** Model study sites: (**a**) Proctor Creek, Atlanta, GA, USA; (**b**) Bronx Wash, Tucson, AZ, USA.

**Table 1.** Study site characteristics.

| | Drainage Area (km$^2$) | Mean Basin Slope (%) | % Imperviousness | % Developed |
|---|---|---|---|---|
| Bronx Wash | 3.2 | 1.89 | 46 | 100 |
| Proctor Creek | 42 | 9.76 | 35 | 84 |

Bronx Wash is characterized by a semi-arid climate receiving 300 mm of rain annually. Precipitation is bimodal with approximately half the annual precipitation occurring during the summer months as monsoonal rainfall characterized by short-duration, high-intensity events with high spatial heterogeneity. Winter events tend to have longer duration, lower intensities, and larger spatial extents. Proctor Creek experiences a similar bimodal pattern; however, it is characterized by a humid subtropical climate receiving approximately 1195 mm of rainfall annually. Mixed deciduous forest and vegetated open space typically characterize undeveloped portions of Proctor Creek. Undeveloped areas of Bronx Wash predominantly consist of herbaceous patches and bare soil. This results in higher fractions of rainfall to runoff and a decreased lag to peak time compared to Proctor Creek. Consequently, the uncertainty in model inputs and parameters for each site are different (Table 2).

*2.2. Hydraulic Model*

Model simulations of flood hydraulics were conducted with the Hydrologic Engineering Center's River Analysis System 5.0.7 (HEC-RAS). HEC-RAS is capable of simulating steady and unsteady flows and 1-D and 2-D hydraulics. Considering sources of uncertainty in addition to flood magnitude, such as channel change and land use, greatly increases the complexity and computational demand of conducting MCS in a 2-D analysis compared to 1-D. Therefore, we conducted a steady, 1-D flow analysis to provide a cost-effective method of simulating uncertainty in land use and channel change with MCS.

**Table 2.** Sample parameterization for MCS.

| Bronx Wash | Distribution | μ | σ |
|---|---|---|---|
| Flood quantile 0.99 (cms) | Lognormal | 16.9 | 4.6 |
| Flood quantile 0.98 (cms) | Lognormal | 13.2 | 3.7 |
| Manning's n-value (channel) | Lognormal | 0.017–0.035 | 0.09–0.12 |
| Manning's n-value (floodplain) | Lognormal | 0.025–0.055 | 0.11–0.15 |
| Channel change (m) | Normal | 0 | 0.61 |
| Proctor Creek | | | |
| Flood quantile 0.99 (cms) | Lognormal | 221.8 | 33.3 |
| Flood quantile 0.98 (cms) | Lognormal | 189.4 | 23.9 |
| Manning's n-value (channel) | Lognormal | 0.04–0.065 | 0.13–0.15 |
| Manning's n-value (floodplain) | Lognormal | 0.07–0.11 | 0.16–0.18 |
| Channel change (m) | Normal | 0 | 0.09 |

HEC-RAS solves the energy equation using the standard step method in a 1-D steady flow analysis (Equation (5)).

$$Z_2 + Y_2 + \frac{\alpha_2 V_2{}^2}{2g} = Z_1 + Y_1 + \frac{\alpha_1 V_1{}^2}{2g} + h_e \tag{5}$$

where Subscripts 1 and 2 identify upstream and downstream cross sections, respectively, $Z$ is the elevation of the channel thalweg, $Y$ is the water surface elevation, $V$ is the cross-sectional average velocity, $\alpha$ is a weighting coefficient, $g$ is gravitational acceleration, and $h_e$ is the energy loss between Cross Sections 1 and 2. The specific energy at a cross section is calculated as

$$E = Z + Y + \frac{\alpha V^2}{2g} \tag{6}$$

The energy loss, $h_e$, is a function of the distance between the two cross sections, the expansion and contraction loss parameters, and the friction slope between the two cross-sections, $S_f$:

$$S_f = \left[\frac{Q}{K}\right]^2 \tag{7}$$

$$K = \frac{1}{n} A R^{2/3} \tag{8}$$

where $Q$ is the volumetric water discharge, $K$ is conveyance to allow subdivision of cross-sections, $n$ is Manning's roughness coefficient, $A$ is the flow area, and $R$ is the hydraulic radius.

The implemented models were based on regulatory floodplain models, which were developed, calibrated, and approved by professional engineers to regulate floodplain development. The regulatory floodplain model at Proctor Creek was developed in a recent version of HEC-RAS (5.0.4). Therefore, the existing calibration was assumed to be adequate, and its parameters were taken as the mean condition in MCS. The regulatory model at Bronx Wash was developed in HEC-2, the precursor to HEC-RAS. Therefore, it was converted to HEC-RAS and calibrated to observed water depths during the summer monsoon season of 2017. Calibration at Bronx Wash matched observations within 0.06 m and was achieved by adjusting channel n-values and expansion/contraction coefficients.

*2.3. Monte Carlo Simulations*

MCS was conducted according the probabilistic floodplain mapping framework of [3], and 1000 simulations achieved statistical convergence. For each simulation, latin-hypercube sampling populated a parameter set of discharge, channel and floodplain roughness values, and channel bed elevation. The MCS generated 1000 different flood scenarios each with a

unique water surface. Each water surface was compared with a 1 m digital elevation model (dem) to delineate 1000 unique inundation areas, which quantified the spatial distribution of inundation probability for a flood of a specified occurrence likelihood, such as the 1% AEP flood (Figure 2). The non-exceedance inundation probability for a particular location or pixel on a floodplain was determined by

$$P_{i,Q} = 1 - \frac{1}{n} \sum_{j=1}^{n} f_{i,j} \tag{9}$$

where $i$ indicates the pixel or location on the floodplain, $Q$ denotes the flood of specified occurrence probability, $n$ is the number of simulations, and $f_{i,j}$ is the inundation status (1 = wet, 0 = dry) at a pixel for simulation $j$ of $n$.

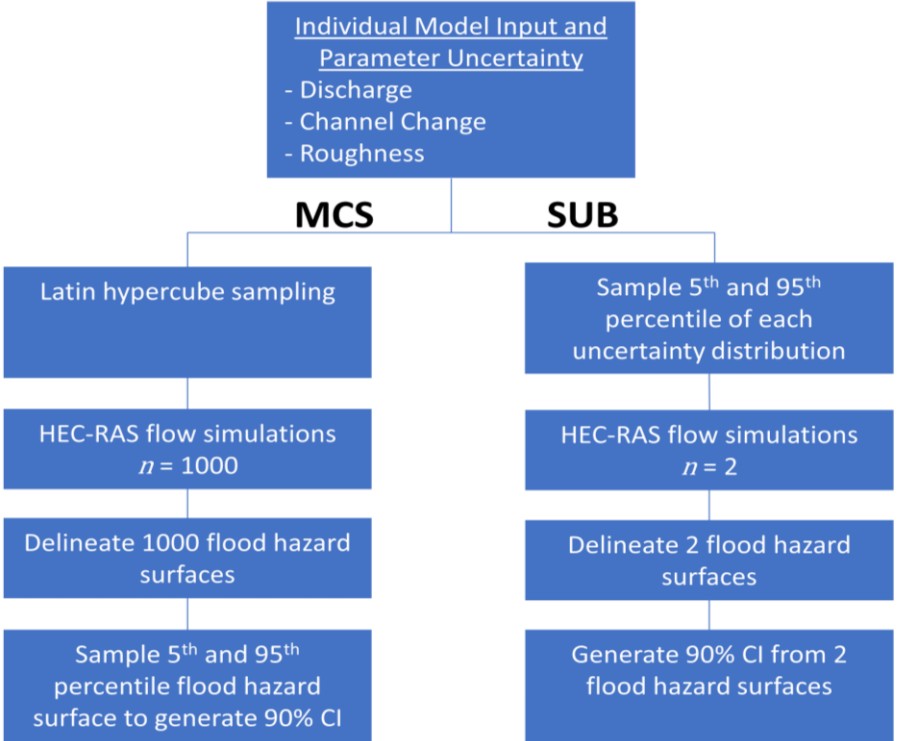

**Figure 2.** Flow chart describing the process to create a 90% confidence interval (CI) of flood hazard estimates using MCS and the SUB method.

The Hydrologic Engineering Center's Statistical Software Package (HEC-SSP) quantified uncertainty in discharge at Proctor Creek by estimating the magnitude–frequency relationship with a Log-Pearson III distribution and annual peak flood observations at the United States Geological Survey stream gage 02336526 from 1961 to 2018; however, gaps in the observational record reduced the number of annual peak flood observations to 25, increasing the standard deviation in quantile estimates. A regional flood regression equation determined the magnitude–frequency relationship at Bronx Wash, since stream gage observations did not exist in the watershed [37]. A two-parameter log-normal distribution defined uncertainty in discharge quantile estimates such that the HEC-SSP quantified the standard deviation at Proctor Creek, and the standard error of prediction in the regional regression equation defined the standard deviation at Bronx Wash.

Performance of SUB and factors impacting differences between the two approaches were evaluated for two separate flood frequencies, the 1% and 2% annual exceedance probability (AEP) flood (100-year and 50-year flood). These frequencies were selected because they are both socially and practically relevant in terms of floodplain management

and infrastructure design but evaluate performance for two separate points along the flood magnitude–frequency curve.

The existing Manning's n-values in the calibrated regulatory model quantified the mean roughness values. The distribution was assumed to follow a two-parameter lognormal distribution [32,38], and Equation (10) from Burnham and Davis [38] quantifies the standard deviation:

$$\sigma = Nr * (0.582 + 0.1 * ln(n)) \tag{10}$$

where *Nr* is a reliability constant ranging from 0 to 1 based on the confidence in the estimated Manning's *n*-value (0 = perfect confidence, 0.5 = moderate confidence). A value of 0.5 was used to represent moderate confidence in this analysis. From this approach, a unique mean and standard deviation described each prescription of Manning's *n*-value within a cross section.

Evaluation of manual field measures at the USGS stream gage 02336526 quantified uncertainty in the channel bed elevation at Proctor Creek [3,39]. Historic measurements of channel bed elevation change at two adjacent rivers were used to bracket the response of channel change at Bronx Wash [40,41]. A normal distribution described channel change, which was simulated by uniformly adjusting the channel bed up or down depending on the sampled parameter set. Sections of the channel at Bronx Wash are composed of concrete. Degradation was not simulated at these locations; however, aggradation was simulated. In other locations, Bronx Wash is characterized by a sand bed channel with a high stream response potential due to channel resistance relative to the magnitude and frequency of expected flows. As such, an increased mean and standard deviation described the distribution of channel change at Bronx Wash compared to Proctor Creek, since the channel at Proctor Creek is characterized by a mixed sand and gravel bed with vegetated banks. However, the anthropogenic impacts on cross-sectional shape, planform, and floodplain topography are evident at both locations, as is typical of many urban stream corridors.

## 2.4. The Simplified Point Approach

We developed the novel SUB method as an alternative to MCS for estimating uncertainty in flood hazard estimates. It involves conducting deterministic model simulations where select quantiles of the model input and parameter distributions are systematically sampled. For instance, a 90% confidence interval on inundation extents from SUB would be generated by running two deterministic simulations that implemented the 5th and 95th percentile of each model input and parameter distribution (Figure 2). Thus, one could evaluate a desired range of uncertainty by conducting two deterministic model simulations that bracket an upper and lower bound based on quantified distributions of model inputs and parameters.

## 2.5. Evaluation Metrics

The uncertainty in flood hazard estimates derived from MCS served as a reference to evaluate the accuracy of SUB by comparing flood area, depth, and width. To make this comparison, systematically sampled quantiles of the SUB method were compared against equal quantiles of non-exceedance probabilities from MCS results. For example, the inundation area from a systematic sample of the 5th percentile of the input distributions in the SUB method was compared against the flood area that was not exceeded in 5% of the MCS.

The critical success index (CSI) (Equation (11)) was used to evaluate the inundation area. The CSI is commonly used to evaluate the performance of a simulation relative to a validation area that is effective in floodplain mapping when the focus is on the spatial distribution of flood extents [42,43].

$$CSI = \frac{A}{A + B + C} \tag{11}$$

where $A$ = Hit—a validation area correctly identified as inundated by the simplified approach; $B$ = False alarm—an incorrectly identified inundated area by the simplified approach outside the validation area; $C$ = Miss—a validation area identified as dry by the simplified approach. In addition to the CSI, total inundated area estimated by SUB and MCS were compared as a ratio (i.e., SUB inundated area divided by MCS inundated area).

Relative error normalizes the difference between an estimate and an exact value by the exact value. While the CSI provides a measure of aerial accuracy, differences in the relative error of flood depth and top width provide a measure that is directly relatable to acceptable or inacceptable levels of error in additional flood hazard metrics (e.g., ±10% in depth). Consequently, the relative error in flood depth and top width at model cross sections were also compared between the two approaches. The hydraulic depth in the channel of MCS quantiles normalized absolute depth errors, while the top width of MCS quantiles normalized the absolute width error. Quantifying depth, width, and area provide a comparison of different hazard types (i.e., extent and intensity), which is important for determining where their uncertainty distributions might be inversely related due to valley shape.

### 2.6. Impact of Hydraulic Structures

The impact of hydraulic structures on the performance of SUB was evaluated by removing structures from the model and re-analyzing the performance of SUB. Errors with and without structures provided a means to isolate their impact. The absolute value of relative errors for simulations with structures were subtracted from the absolute value of simulations without structures. In this case, positive values indicated that structures increase error, and negative values indicated that structures decrease error (i.e., more closely match MCS results). A significant portion of Bronx Wash flows through a culvert under a parking lot, and removing this structure from the model would not provide a meaningful comparison. The hydraulic structures at Proctor Creek primarily consist of bridges and roadway culverts of relatively short length. Therefore, the impact of hydraulic structures was only investigated at Proctor Creek.

### 2.7. Correlation Metrics

The performance of SUB might vary due to differences in input and parameter distributions; however, physical characteristics that control the rate of hydraulic response might also impact its performance. We used ordinary least squares regression to evaluate the correlation between multiple physical and hydraulic covariates with SUB errors. Covariates were selected based on their ability to describe morphometric or hydraulic attributes that might influence the sensitivity of uncertainty, including confinement ratio, friction slope, and Froude number.

Confinement ratio was calculated as the ratio of the 100-year floodplain width to the 2-year floodplain width and served as a metric to characterize valley shape. A lower value of this metric describes a narrower, confined valley, while higher values represent valleys with wider floodplains relative to channel width. In confined valleys, we anticipate that flood depth will exhibit greater sensitivity to changing inputs and parameters and therefore greater error in SUB; in less confined valleys, we anticipate a greater sensitivity of flood width.

Friction slope served as a hydraulic metric to describe the impacts of various physical characteristics and their interacting effects on hydraulics, such as changes in bed slope, valley constrictions, and spatially varying roughness coefficients. These characteristics can have a substantial impact on the flow but are difficult to isolate, and friction slope reflects these impacts.

The Froude number quantifies the ratio of gravitational forces to inertial forces, indicating a departure from the minimum specific energy and subcritical versus supercritical flow conditions. It was selected as a covariate because the rate of change in water surface

elevation relative to specific energy varies with the departure from critical flow conditions and the minimum specific energy.

Ordinary least squares regression in the R statistical program was used to quantify the relationship between covariates and SUB errors. In some instances, natural log-transformed values served as a more appropriate model, which was determined based on visual inspection and the quantitative range of the variables in question. Regression diagnostics were evaluated based on the coefficient magnitude, *p*-value, and visual inspection [44]. Further, Spearman's ρ provided a non-parametric evaluation, resistant to outliers, of the monotonic relationship between the SUB error and the co-variates [44]. Error at individual quantiles constituted the sample population in the regression analysis, and the unique covariates for each sample were based on the results of SUB.

## 3. Results

### 3.1. Inundated Area

The 90% confidence intervals in inundation extents estimated by SUB generally agreed with MCS (Figure 3). However, the accuracy of the SUB method was spatially variable. Noticeable deviations occurred in certain locations, while the results appeared identical to MCS in other locations. Deviations at the 95% quantile tend to overestimate the results of MCS, and they tend to underestimate the results of MCS at the 5% quantile, presenting a more precautionary estimate of uncertainty and flood risk.

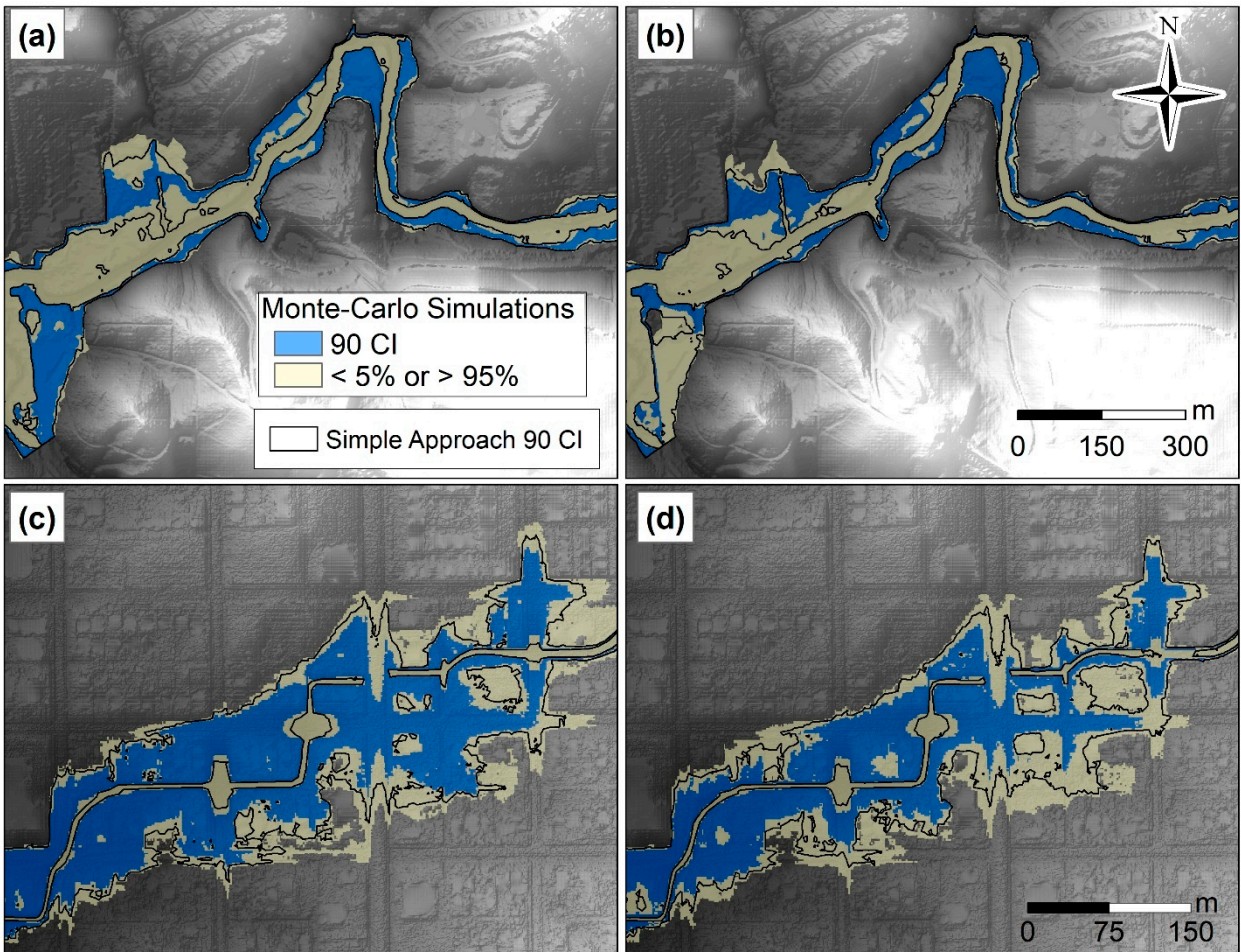

**Figure 3.** 90% confidence intervals (CI) of flood inundation estimated by MCS and the SUB for (**a**) the 1% AEP flood at Proctor Creek, (**b**) the 2% AEP flood at Proctor Creek, (**c**) the 1% AEP probability flood at Bronx Wash, and (**d**) the 2% AEP flood at Bronx Wash. The 90% CI indicate areas that have a 5–95% chance of being inundated by the flood with a specific AEP.

The spatial patterns of inundation among the 1% and 2% AEP flood magnitudes appeared similar, but the uncertainty appeared greater for the 1% AEP flood compared to the 2% AEP for both Proctor Creek and Bronx Wash. Additionally, differences in inundation extent accuracy emerged between the two flood magnitudes. For instance, SUB of the 2% AEP flood at Bronx Wash deviated from MCS to a greater extent along the southeast inundation boundary compared to the 1% AEP (Figure 3c,d). Thus, the performance of SUB varies spatially and among different flood magnitudes.

The CSI and percentage of inundated area indicate that SUB simulates the mean condition with high accuracy, but performance declines with increasing distance from the mean (Figure 4). Performance appeared to be worse at the lowest quantiles (less than 0.05). Despite this relationship, the CSI was greater than 80% for all quantiles exceeding 0.1 at Proctor Creek and greater than 70% for all quantiles exceeding 0.05 at Bronx Wash. Lower quantiles resulted in an underestimation of inundated area, and higher quantiles resulted in an overestimation of inundated area. This general error pattern overestimates the uncertainty quantified by MCS, supporting observations in Figure 3.

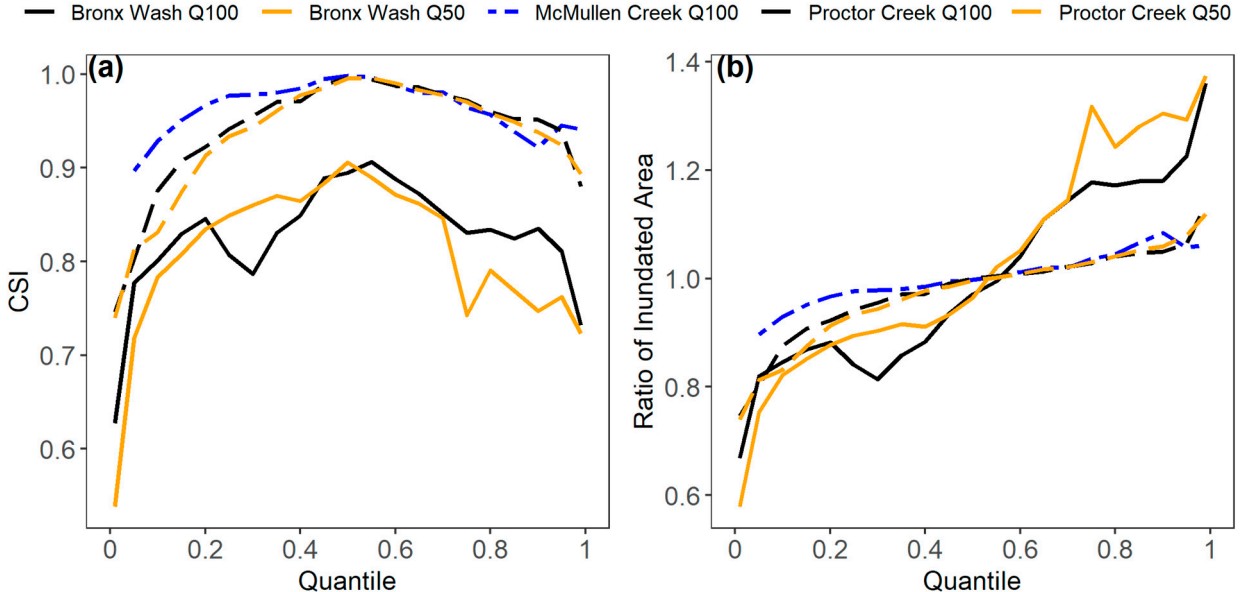

**Figure 4.** (**a**) Critical success index comparing SUB against MCS for each reach and flood frequency and (**b**) the ratio of inundated area estimated by SUB relative to MCS. McMullen Creek from Stephens and Bledsoe [3].

For comparison, we included the results of Stephens and Bledsoe [3] at McMullen Creek, Charlotte, NC. Performance was improved at Proctor Creek and McMullen Creek compared to Bronx Wash, which is likely due in part to the reduced variance in model input and parameter uncertainty compared to Bronx Wash. In general, performance among flood frequencies at each reach was similar with the exception of a minor deviation around the 0.8–0.95 quantile at Bronx Wash. This result contrasts the spatial variation observed in Figure 3, suggesting that spatial patterns in error shift with discharge so that performance is similar among flood magnitudes; however, spatial variation in performance occurs due to variations in local channel and floodplain characteristics that propagate uncertainty differently.

### 3.2. Relative Error in Depth and Top Width

Relative depth errors in SUB varied among quantiles and locations (Figure 5). Similar to the CSI, quantiles closest to the median generally resulted in the least amount of error for Proctor Creek. However, a larger number of points close to the median quantile deviated from the error of 0 at Bronx Wash, which supports its reduced CSI at the median quantile compared to Proctor Creek. Errors at individual stations matched the overall patterns

depicted by the CSI at Proctor Creek. Higher quantiles tended to overestimate depth, and lower quantiles tended to underestimate depth. Error patterns among the two flood magnitudes were similar for Proctor Creek and Bronx Wash.

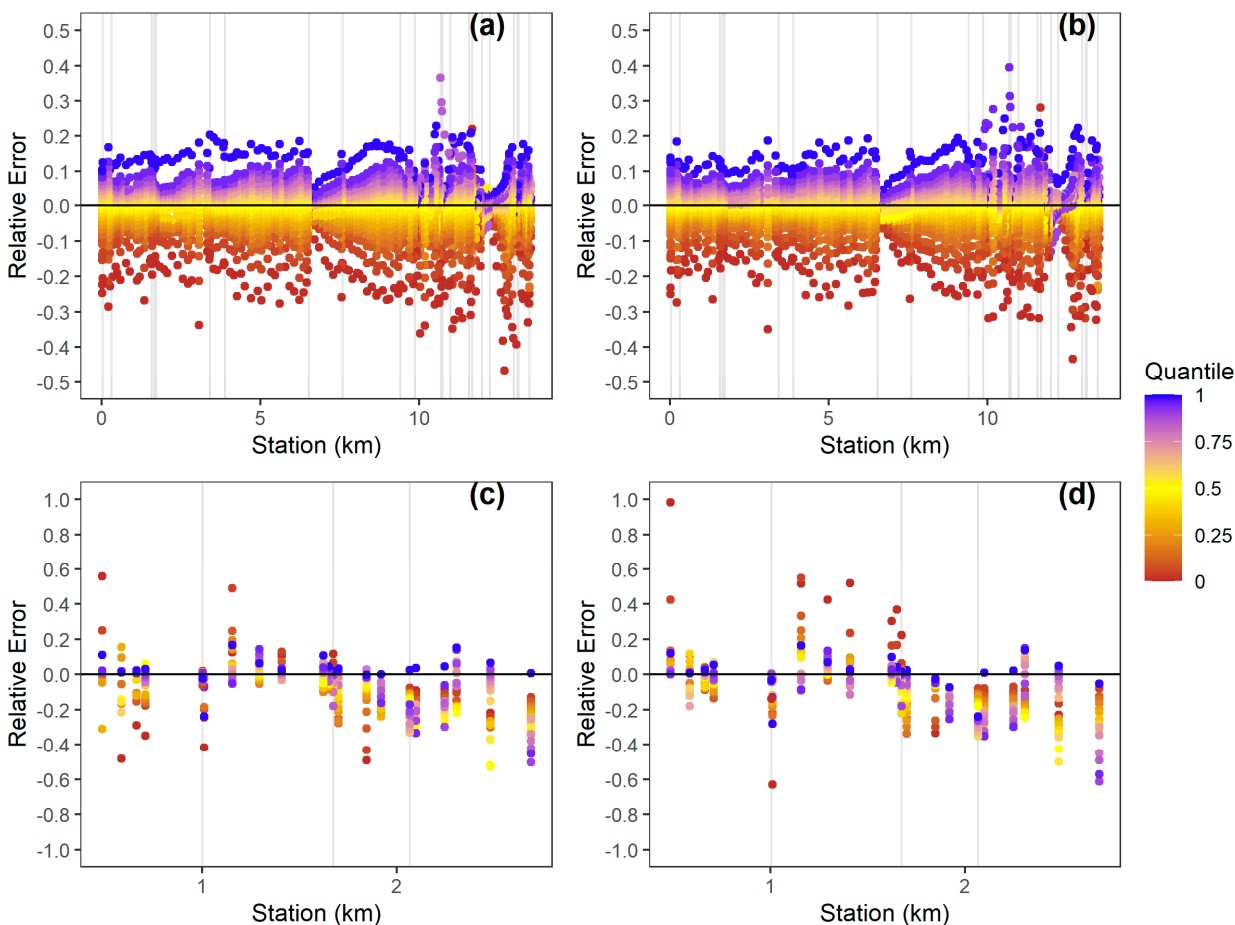

**Figure 5.** Relative depth errors in SUB relative to MCS for (**a**) the 1% AEP flood at Proctor Creek, (**b**) the 2% AEP flood at Proctor Creek, (**c**) the 1% AEP probability flood at Bronx Wash, and (**d**) the 2% AEP flood at Bronx Wash. Vertical gray lines indicate hydraulic structure locations.

With a few exceptions, relative errors of less than ±0.2 generally characterized high quantiles, and quantiles around 80–90% tended to approximate MCS with a relative error of less than ±0.1. Based on these results, SUB could estimate the upper 90% confidence limit of depth with a relative error of approximately 0.1. Some locations would have smaller errors, and a few outliers would have larger errors. Lower quantiles, especially those below 5%, generated the largest error magnitudes at Proctor Creek, as evidenced by the proportion of points exceeding a relative error of ±0.2. However, the error among stations exhibited much more variability at Bronx Wash, and a clear relationship between relative error and quantiles was not visible.

Negative values described the relative depth errors for stations greater than 2 km at Bronx Wash. That section of the wash has a concrete channel boundary, and the error is likely influenced by lower n-values and a truncated channel change distribution that did not simulate degradation. Relative depth errors at Bronx Wash tended to be greater than Proctor Creek, and several outliers elevate the relative depth error to more than ±0.4.

A rapid decline in relative depth error located near the 6 km station at Proctor Creek appeared to coincide with a structure location, as indicated by the vertical gray lines, suggesting that the structure imposes a hydraulic control and impacts the accuracy of SUB. A similar pattern where relative errors decrease appears at other structure locations. However, an obvious relationship between relative depth error was difficult to decipher

at all structure locations due to their abundant presence. A relationship between relative depth error and structure locations did not readily present itself at Bronx Wash.

The relative top width error of the simplified approach exhibited patterns similar to those of the relative depth error; however, many more outliers exist (Figure 6). Bronx Wash exhibited larger magnitudes of relative depth error, and Proctor Creek exhibited larger magnitudes of relative top width error. Larger quantiles generally resulted in relative top width outliers at Proctor Creek, but low and high quantiles resulted in relative top width error outliers at Bronx Wash. According to Figure 6, the simplified approach can be expected to result in a relative error in top width of approximately 0.1–0.2, with the exception of some outliers that might be higher or lower. More extreme outliers existed at high quantiles, with rare instances of elevated relative top width errors ranging from 1 to nearly 4. However, outliers at low quantiles did not exceed 1.

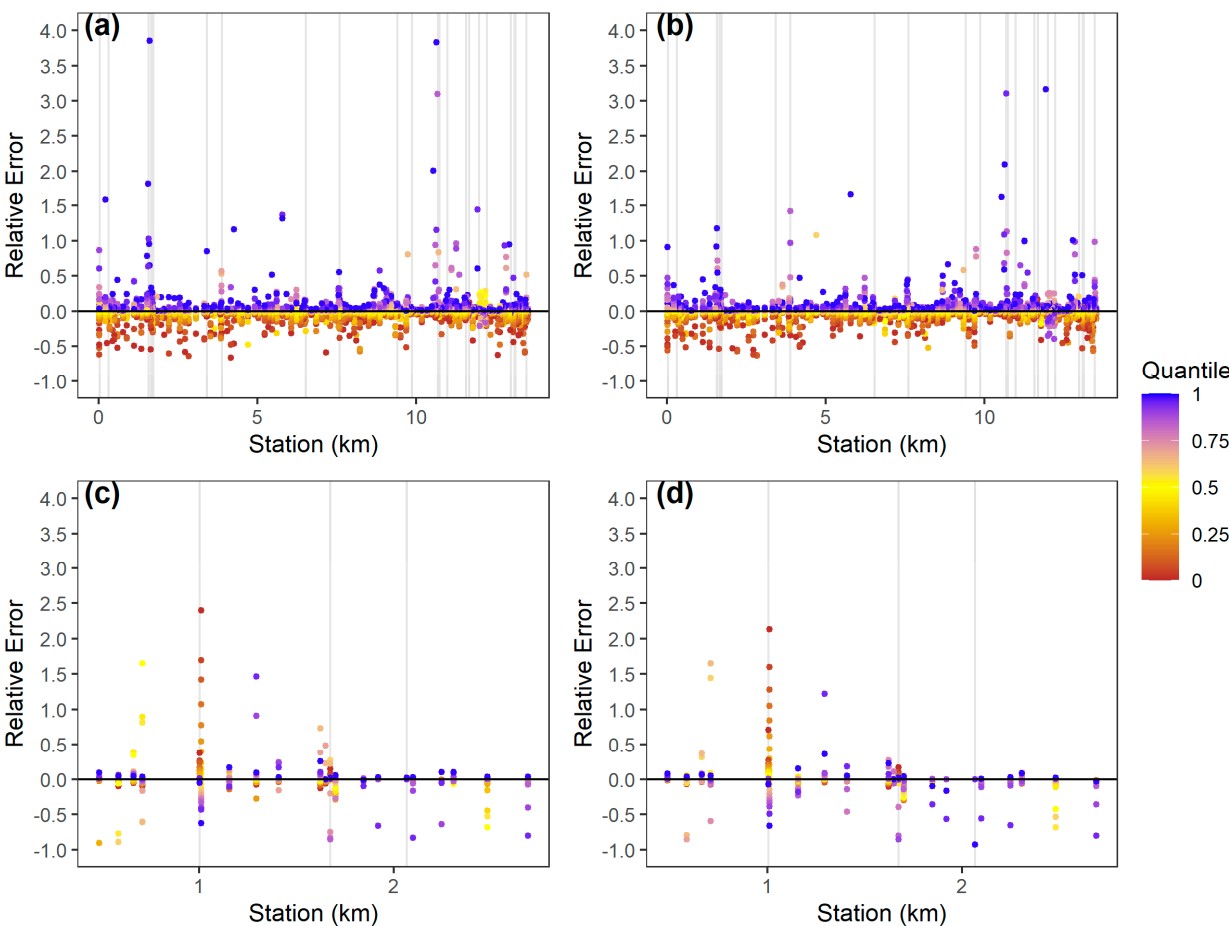

**Figure 6.** Inundated top width relative errors in SUB relative to MCS for (**a**) the 1% AEP flood at Proctor Creek, (**b**) the 2% AEP flood at Proctor Creek, (**c**) the 1% AEP probability flood at Bronx Wash, and (**d**) the 2% AEP flood at Bronx Wash. Vertical gray lines indicate hydraulic structure locations.

While relative errors provide a dimensionless evaluation that is comparable among locations, absolute error magnitudes aid in evaluating SUB. Even though a greater variance in relative depth error occurred at Bronx Wash, greater variance existed in the absolute magnitude of depth error at Proctor Creek (Figure 7). However, the distribution of top width error was nearly identical among locations, with the exception of large positive errors occurring above the 95% quantile. Significant differences in the distribution of absolute error among flow magnitudes did not exist at either location. The cumulative distribution of absolute errors highlights the general accuracy that can be expected by SUB. For example, SUB could generally be expected to approximate the upper or lower 90% confidence limits

of inundation width with ±20 m accuracy, noting the exception of some outliers that would have larger errors.

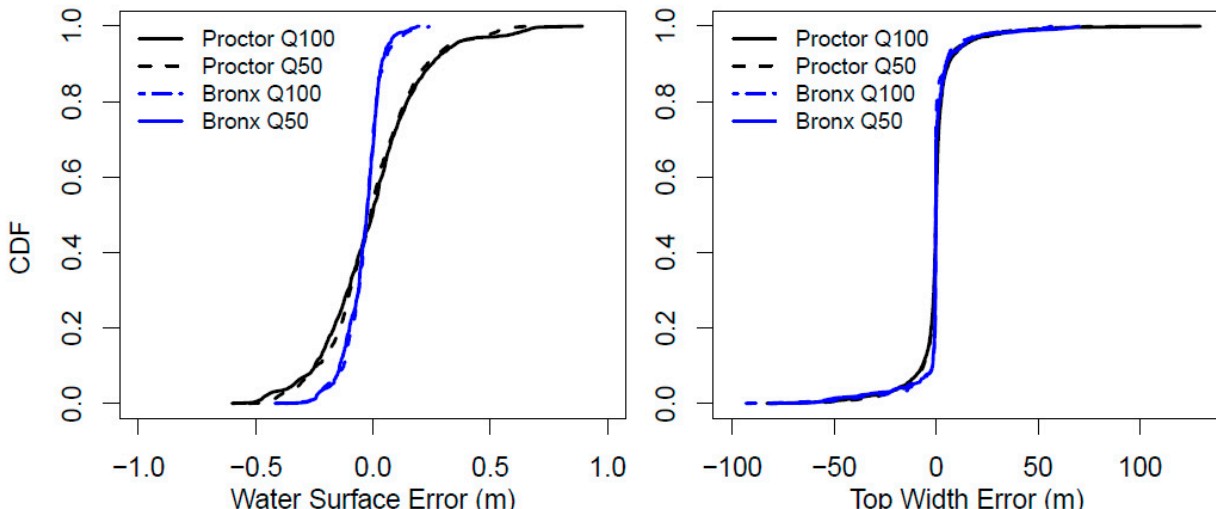

**Figure 7.** Cumulative distribution of the absolute depth (**left**) and inundated top width (**right**) errors in SUB relative to MCS.

### 3.3. Hydraulic Structures

Hydraulic structures had a heterogeneous impact on relative error at Proctor Creek (Figure 8). Negative values in Figure 8 indicated instances where structures reduced error, and positive values indicate instances where structures increased error in SUB. The impact of structures varied among quantiles, flow magnitudes, and stations. However, it is important to note that the impact of hydraulic structures on the accuracy of SUB propagated upstream and downstream, but the impact diminished with increasing distance from the structure. Structures tended to increase the relative depth error for lower quantiles of the 2% AEP flood at Proctor Creek and decreased error for higher quantiles. Thresholds might exist at structures or topographic transitions where hydraulic conditions change rapidly with increasing/decreasing water levels. For instance, a bridge may reduce errors by constricting flow and partially controlling the maximum flow rate on the downstream side of the bridge once water levels reach the bottom of the bridge deck but before overtopping occurs. However, the impact of hydraulic structures on the accuracy of SUB was neither negligible nor straightforward.

### 3.4. Correlation Metrics

The ability to predict where SUB errors might influence uncertainty analyses could aid in the inference of results or in determining its applicability. We evaluated three hydraulic metrics across 21 quantiles and two AEPs at each site to identify variables that might predict an increase or decrease in depth or top width errors from SUB. This resulted in a total of 126 correlation plots. For organizational purposes, regression diagnostics are available in the Supplementary Materials. Here, we summarize the general findings from the regression and trend analysis and highlight the significance of each metric to predict error magnitudes of SUB (Tables 3 and 4). Tables 3 and 4 provide a qualitative description. The following paragraphs provide a more quantitative summary of the analysis.

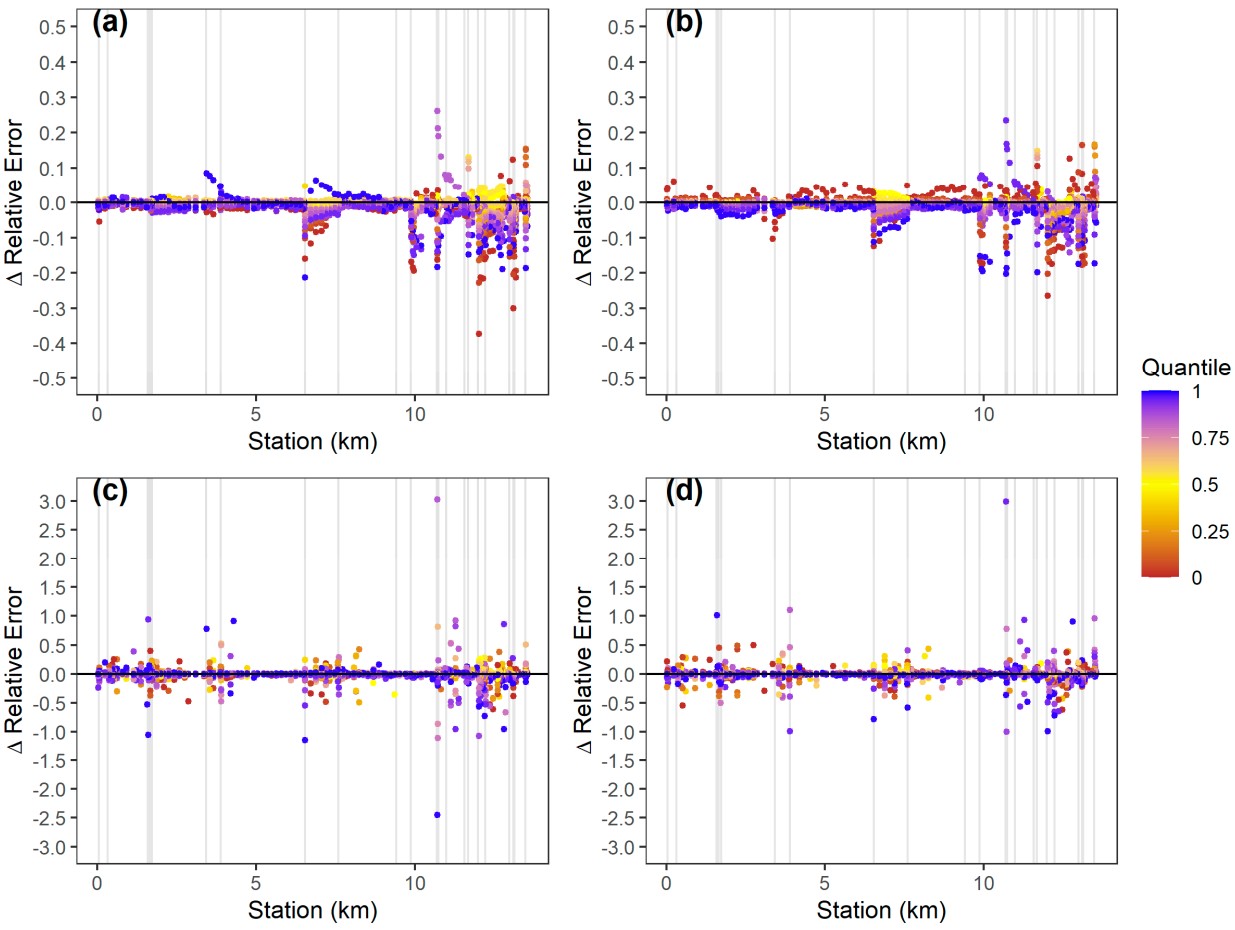

**Figure 8.** Difference in the absolute value of relative error between simulations of SUB with structures and without structures for the (**a**) 1% AEP depth, (**b**) 2% AEP depth, (**c**) 1% AEP top width, and (**d**) 2% AEP top width.

**Table 3.** General trends between top width errors in SUB and physical and hydraulic metrics. (+) indicates a positive trend, (−) indicates a negative trend, and (0) indicates no apparent trend.

| Predictor | Proctor Creek | | Bronx Wash | |
|---|---|---|---|---|
| | **1% AEP** | **2% AEP** | **1% AEP** | **2% AEP** |
| Confinement Ratio | + | + | + | + |
| Froude Number | − | − | 0 | 0 |
| Friction Slope | − | 0 | 0 | 0 |

**Table 4.** General trends between depth errors in SUB and physical and hydraulic metrics. (+) indicates a positive trend, (−) indicates a negative trend, and (0) indicates no apparent trend.

| Predictor | Proctor Creek Q | | Bronx Wash | |
|---|---|---|---|---|
| | **1% AEP** | **2% AEP** | **1% AEP** | **2% AEP** |
| Confinement Ratio | 0 | 0 | 0 | 0 |
| Froude Number | 0 | 0 | 0 | 0 |
| Friction Slope | − | − | + | + |

Overall, a positive relationship was identified between the top width error and the confinement ratio. OLS regression and Spearman's ρ indicated a statistically significant

increase in top width error with an increasing confinement ratio at the 5% significance level. This means that, as the valley width increases relative to channel size, errors in the uncertainty of inundation width will increase. The SUB method provides a more accurate prediction of top width uncertainty where confinement is greater. The rate of change in top width with water surface elevation will typically be greater in the valley bottom compared to the valley walls. Consequently, a larger valley bottom provides a larger area for errors in top width to propagate at a faster rate compared to more confined valleys where top width error is reduced, because the inundation extents reach the valley walls at lower flood depths. Further, the floodplain is often characterized by higher roughness values with greater variance compared to the channel. An increased portion of flow in areas with higher roughness variance will inevitably impact the error of the simple approach.

At Proctor Creek, top width error generally decreased as the Froude number increased; however, the trend was not significant at a 5% significance level for quantiles greater than 70%. A decreasing but insignificant trend was also identified for quantiles less than 50% at Bronx Wash. Above the median, the slope of the trend and correlation coefficient was insignificant and varied among quantiles. A decrease in top width error with an increasing Froude number would suggest that, as the water surface approaches a critical flow depth, the sensitivity of uncertainty to changes in inputs and parameters decreases.

Similar to the Froude number, top width error generally decreased as friction slope increased at Proctor Creek. Trends were statistically significant according to OLS regression and Spearman's $\rho$ for all quantiles less than 95%. Additionally, the magnitude of the trend slope generally decreased with increasing quantiles, indicating the diminishing correlation between friction slope and top width error with increasing top width. However, at Bronx Wash, trend slopes were not statistically different from 0, and an apparent trend was not discernable.

Linear regression and Spearman's $\rho$ did not indicate a correlation between depth error and confinement ratio at Proctor Creek. At Bronx Wash, Spearman's $\rho$ identified a statistically significant negative trend at 8 of 21 quantiles for the 1% AEP. All trend slopes above the 30% quantile were negative, and all trend slopes below the 30% quantile were positive. However, the OLS regression did not support this finding as most trend slopes were insignificant and nearly 0. Visual inspection did not reveal an apparent trend. Consequently, confinement ratio did not appear to be correlated with depth errors. Similarly, trend slopes between the Froude number and the depth error were nearly 0 and insignificant, indicating that the Froude number was a poor predictor of depth errors.

A predominantly statistically insignificant but negative trend was identified between friction slope and depth error at Proctor Creek. Trend slopes were statistically significant at less than half of the quantiles, and with the exception of one quantile, all trend slopes were negative according to OLS regression and Spearman's $\rho$. In contrast, trend slopes between friction slope and depth error were predominantly positive at Bronx Wash with a large majority of the trends identified as statistically insignificant.

## 4. Discussion

The accuracy of SUB in simulating the uncertainty estimated by MCS was variable among the analyzed stream reaches, model cross sections, quantiles, and flood frequencies. Spatial variation in the accuracy of the SUB method is supported by the findings of an FOA analysis at two reaches with different valley shapes [29]. In that analysis, uncertainty was more sensitive to different parameters and propagated differently among reaches with a V-shaped valley and a U-shaped valley. These findings indicate that changes in valley shape and local characteristics that influence model parameterization impact the sensitivity to uncertainty and thus the accuracy of SUB.

The SUB method performed relatively well with the exception of some outliers. Relative errors identified in our analysis are slightly larger than an evaluation of relative error in the uncertainty of levee flow capacity with FOA [20]. They found a relative error of 0.02 and 0.04 in the mean and standard deviation, respectively. Considering that 1-standard

deviation would relate to an upper quantile of 84% for SUB, relative depth errors were generally less than 0.1 at this quantile but variable among stations. Tyagi and Haan [20] quantified relative error in discharge at a single cross section determined by a simplified flow model, and they used different uncertain parameters compared to this analysis. Consequently, an equal comparison of relative errors among studies is difficult. For instance, there are stations in our analysis with relative errors smaller than those quantified by Tyagi and Haan [20], but there were also stations with larger relative errors.

We were unable to identify studies that evaluated flood depth, top width, or inundation area with other point estimation techniques for comparison. However, similar to our results, point estimation techniques were shown to perform adequately but slightly less accurately than FOA methods when calculating the risk of culvert failure [45]. However, there are contrasting results where point estimation techniques did not adequately match the variance of flood hydrographs from MCS in a distributed rainfall–runoff model [46]. Differences in the performance of the SUB method among various flood hazard metrics are similar to a study finding that flood peak and flood volume impacted uncertainty in a hydrologic model differently [47].

The increased relative error at Bronx Wash compared to Proctor Creek is supported by the results of an FOA analysis that shows increasing error with increasing variation in model parameters [20], since model inputs and parameters at Bronx Wash are characterized by higher variance compared to Proctor Creek. Further, decreasing accuracy of SUB with increasing departure from the mean is similar to other studies that evaluate methods for approximating uncertainty [34,48]. However, SUB is straightforward and provides an attractive alternative to other approximation methods that are more complex or require prior knowledge of response functions.

The SUB method generally provides a conservative estimate of uncertainty by overestimating flood hazards at higher quantiles and underestimating flood hazards at lower quantiles. The consequences of overestimating uncertainty in inundation extents might result in increased capital costs; however, the consequences of underestimating uncertainty might result in flood loss or human endangerment [49].

The acceptability of the absolute error magnitudes from SUB will inevitably depend on the specific application. The SUB method nonetheless provides a cost-effective approximation when evaluating the degree of uncertainty. For instance, information gained by SUB can inform investment decisions regarding data collection and analysis approaches. If the uncertainty bounds from SUB are large, more complex approaches such as MCS might be warranted, whereas more certain outcomes might not warrant the added investment. However, it is important to consider that MCS is also an estimation technique that does not provide a perfect quantification of uncertainty [20]. Although the distribution of results obtained from MCS appeared to converge with fewer than 1000 simulations, it is possible that more simulations could be required to adequately sample the extremes in the tails of a distribution.

The lack of readily available data that directly quantify uncertainty distributions of model inputs and parameters, such as channel capacity, presents a limitation for both the SUB method and MCS in practical applications. However, this challenge can be overcome by using regionally and locally available expertise and judgement to reasonably bracket uncertainty distributions. Another limitation exists in the fact that there is uncertainty in the functional distributions applied to estimate extreme values and uncertainty itself (e.g., Log Pearson Type III or Generalized Extreme Values distributions).

Hydraulic structures had significant but heterogeneous impacts on SUB errors. However, the model structure could impact these results due to shifts in the numerical scheme, when flow through a bridge or culvert becomes pressurized and transitions to overtopping. Future studies may seek to perform similar analyses with additional models that utilize different numerical schemes at hydraulic structures. However, HEC-RAS is a widely accepted and readily available 1-D hydraulic model for simulating flood hazards. The accuracy of the numerical scheme to simulate actual flow conditions for a given set of inputs and

parameterizations will dictate the overall level of impact. Despite this added level of uncertainty, the variation in the results of simulations with and without structures highlights the fact that structures present thresholds that can increase or decrease SUB errors.

While errors in SUB were spatially variable, the correlation results indicated scenarios where errors are likely to be greater. This information can help guide application of the approach and interpretation of the results. For example, unconfined flat floodplains will likely result in conservative estimates of uncertainty, considering that top width errors increased as the confinement ratio increased. Varying trends between depth and top width errors with individual hydraulic metrics are supported by the findings of Gates and Al-Zahrani [19]. They found that channel geometry had the smallest impact on flow depth uncertainty compared to bed slope and discharge, but channel geometry had the largest impact on flow velocity. The variation among trends highlights the complex relationship between non-uniform topography and hydraulic conditions.

Evaluating additional metrics may provide further insight on unique features that control uncertainty propagation, and it may enable quantitative adjustments to the SUB to more accurately simulate the uncertainty estimated by MCS. For instance, we utilized the confinement ratio to describe valley shape. However, this metric may fail to capture abrupt topographic shifts, e.g., in a floodplain terrace, where the inundation extent drastically increases beyond a threshold depth. Certain features may have a substantial impact on flood extent uncertainty relative to a threshold depth. An analysis of larger river systems would enhance the transferability of results and, given the likely increase in flood hazard magnitude, may reveal an increase or decrease in the relative error of the SUB method compared to MCS.

## 5. Conclusions

We quantified the uncertainty in inundation area, flood depth, and top width by varying discharge, roughness coefficients, and channel bed elevation in the MCS of flood hydraulics at two reaches with contrasting hydrologic and geomorphic settings. We also quantified uncertainty with the novel SUB method by conducting model simulations based on a systematic sampling scheme of uncertain inputs and parameters. The SUB method was evaluated against MCS through the CSI, the percentage of inundated area, and the relative errors in depth and top width. Further, we conducted regression and trend analysis to identify apparent topographic and hydraulic parameters that might impact the accuracy of SUB. The main conclusions drawn from this analysis are as follows:

1.  The accuracy of SUB to estimate uncertainty from MCS was variable among reaches, spatially within reaches, and across quantiles of the uncertainty distribution. However, accuracy generally increased with decreasing deviation from the mean, and accuracy decreased with increased variance in model inputs and parameters.
2.  The CSI and percentage of inundated area indicated that SUB was highly accurate, but the relative top width error indicated poorer performance, especially for some outliers.
3.  Hydraulic structures can significantly impact the accuracy of SUB but in a non-uniform manner. The direction of the error depends on the quantile and location. This suggests that structures present thresholds in the sensitivity of hydraulic response to uncertainty in model inputs and parameters.
4.  Results of the regression and trend analysis indicated varying influences of hydraulic and topographic metrics, highlighting the complexity of hydraulic processes. However, a positive relationship was consistently identified between the confinement ratio and the top width error, indicating that SUB will overestimate uncertainty in flood width for less confined floodplains. Consistent relationships with additional metrics were not identified among stream reaches.

Overall, SUB performed reasonably well despite some outliers with increased error, but it requires careful consideration of the effects of hydraulic structures and topographic complexity on accuracy. Adequacy of SUB will ultimately depend on the application, but the results of this analysis can be used to inform anticipated error magnitudes and acceptability.

The results of this analysis have described the accuracy of SUB at contrasting settings and highlighted conditions that might reduce accuracy. However, additional research is warranted to provide further insight on the impacts of a model's numerical scheme at hydraulic structures, additional metrics that may impact uncertainty propagation, and transferability to larger river systems. Despite errors relative to MCS, SUB can provide a valuable and cost-effective tool for uncertainty analysis, particularly where resources limit the practicality of MCS. For instance, reducing the computational demands compared to MCS provides accessibility to a larger number of practitioners. Certain applications might require the implementation of more complex uncertainty analyses, but SUB can help inform this decision. Regardless, these results underscore the high degree of inherent uncertainty associated with predicting flood hazards from extreme events.

**Supplementary Materials:** The following supporting information can be downloaded at https://www.mdpi.com/article/10.3390/w14101618/s1. Figures S1–S24 present ordinary least squares and Spearman's Rho trend and correlation significance between SUB errors and all analyzed covariates for each annual exceedance probability simulated.

**Author Contributions:** Conceptualization, T.S. and B.B.; methodology, T.S.; formal analysis, T.S.; investigation, T.S.; resources, B.B. and T.S.; writing—original draft preparation, T.S.; writing—review and editing, B.B.; visualization, T.S.; supervision, B.B.; project administration, B.B.; funding acquisition, B.B. All authors have read and agreed to the published version of the manuscript.

**Funding:** This research was funded by the NSF Sustainability Research Network Cooperative Agreement 1444758, the Urban Water Innovation Network, and by the US Army Corps of Engineers' Engineering with Nature® Initiative through the Cooperative Ecosystem Studies Unit Agreement W912HZ-20-2-0031. Travel for this work was funded by the Consortium of Universities for the Advancement of Hydrologic Science, Inc.'s 2016 Pathfinder Fellowship.

**Data Availability Statement:** All data code generated as a result of this study are available upon request.

**Acknowledgments:** The authors thank Tom Meixner and many others at the University of Arizona for their hospitality and assistance in data collection and field investigations at Bronx Wash.

**Conflicts of Interest:** The authors declare that there is no conflict of interest.

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
