# Peer review of "Simplified Uncertainty Bounding: An Approach for Estimating Flood Hazard Uncertainty"

_water, doi:10.3390/w14101618_

Round 1

Reviewer 1 Report

Dear Authors and Editor,

Thanks for your kind invitation to review the manuscript entitled “Simplified Uncertainty Bounding: An approach for estimating flood hazard uncertainty” by Tim Stephens and Brian Bledsoe.

This study investigates the degree of uncertainty in flood risk estimated using an alternative parsimonious approach compared to Monte-Carlo simulations (MCS). It also identifies the sensitivity of factors and model parameters that influence the uncertainty estimates. I recommend a Major Revision. Here are my suggestions:

  1. It is necessary to consider the computational time step in 2D flow modelling. There is a need to choose a suitable computational time step that works well with the chosen mesh. It must indicate how the value of this parameter has been chosen in the simulations. It may be based on the Courant Number, and applied depending on whether the Saint Venant equation or the Diffuse Wave equation is used to solve the model.
  2. The introduction lacks a general framework for the study. A framing is needed with other related topics and with more bibliographical references. Also the introduction fails to provide a suitable perceptive on the novelty and importance of the study, while this information needs to be stated clearly in the text.
  3. It would be better to highlight major difficulties and challenges and your original achievements to overcome them in a clearer way in the abstract and introduction
  4. In the final part of the introduction you need a brief summary of each section to help you better understand the content of the paper.
  5. The methodology limitation should be mentioned.
  6. The discussion should summarize the manuscript's main finding(s) in the context of the broader scientific literature and address any limitations of the study or results that conflict with other published work.
  7. Perhaps the Conclusions section could be more attractive to the reader if it is presented more concisely. Part of the content is like a summary of part of the article which could be simplified at the beginning and later go fully into the conclusions themselves and in a clear and concise way, perhaps through bullet points.

With kind regards

Author Response

The authors thank the reviewer for the constructive comments. We have accepted or addressed all the comments. A detailed response is provided in the attached document.

Reviewer 2 Report

  • Add more quantified details in the abstract
  • The authors should highlight the problem statement and research significance more clearly in the introduction section.
  • I find that the authors missed some relevant newly published papers in the scope of the current study. Therefore, I suggest including the following references:
  • https://www.hindawi.com/journals/ace/2020/7948902/
  • https://link.springer.com/article/10.1007/s13369-014-1246-9
  •  
  • Please provide a flowchart for describing the approach proposed in the current study
  • The discussion section in the present form is relatively weak and should be strengthened with more details and justifications
  • Please provide the limitations of the study and future directions in the conclusion section.

Author Response

(The authors gave the same response as above.)

Round 2

Reviewer 1 Report

Dear Authors and Editor,

The authors have put diligent efforts in responding to reviewers' comments and they are revised the paper accordingly. The paper is ready. I am happy to recommend acceptance of the paper.

Fine document, congratulations to the authors.

Best regards,

Author Response

The authors thank the Reviewer for their timely and constructive comments that improved the overall quality of the manuscript.

Reviewer 2 Report

Accept

Author Response

(The authors gave the same response as above.)
